# Can the Components of Physical Fitness Be Linked to Creative Thinking and Fluid Intelligence in Spanish Schoolchildren?

**DOI:** 10.3390/healthcare13141682

**Published:** 2025-07-12

**Authors:** Karina Elizabeth Andrade-Lara, Pedro Ángel Latorre Román, Eva Atero Mata, José Carlos Cabrera-Linares, Juan Antonio Párraga Montilla

**Affiliations:** Department of Musical, Plastic and Corporal Expression, University of Jaén, 23071 Jaén, Spain; karinandrade9011@gmail.com (K.E.A.-L.); platorre@ujaen.es (P.Á.L.R.); atero.eva@gmail.com (E.A.M.); jparraga@ujaen.es (J.A.P.M.)

**Keywords:** divergent thinking, physical performance, cardiorespiratory fitness, academic performance, children

## Abstract

**Objective:** The aim of this study was to determine the relationship between the components of physical fitness (PF), creativity and fluid intelligence, as well as to determine which components of PF are predictors of the analysed cognitive potential. **Material and Methods:** A total of 584 Spanish schoolchildren (6−11 years old; age = 8.62 ± 1.77 years) took part in this study. Creativity was assessed using the Torrance Tests of Creative Thinking (TTCT) and fluid intelligence through TEA-1. Moreover, PF components were evaluated using a 25 m sprint, handgrip strength, standing long jump and 20 m SRT. **Results:** Boys exhibited a better PF performance than girls (*p* range from = < 0.001 to 0.05), as well as higher creativity score (*p* < 0.001), the fluid intelligence score and QI score (*p* < 0.05, respectively). Moreover, PF components (CRF, strength and speed) were positively associated with creativity (*p* range from = < 0.001 to 0.001) and fluid intelligence (*p* range from = < 0.001 to 0.015). Regression analysis showed that the creativity model explained between 31.4% and 36.6% of the variance (*R*^2^ = 0.314−0.366, *p* < 0.001), while the fluid intelligence model accounted for 25.5% to 33.1% of the variance (*R*^2^ = 0.255−0.331, *p* < 0.001 to 0.001). **Conclusions:** A positive relationship was found between creativity, fluid intelligence, and PF components. Children with higher PF levels scored better in creativity, with notable differences between boys and girls. These findings highlight the educational value of incorporating structured physical activity into school settings to support both cognitive and physical development.

## 1. Introduction

Regular physical activity (PA) improves physical fitness (PF) by releasing neurotrophic factors, such as brain-derived neurotrophic factor (BDNF), which enhance synaptic formation, neuronal connectivity, and cognitive protection [1]. PA also improves cerebral oxygenation and nutrient supply, optimising brain function and supporting learning processes [2]. Moreover, it stimulates the secretion of key neurotransmitters dopamine, serotonin, and noradrenaline which are involved in emotional regulation, academic performance, and overall well-being in children [3]. 

The main objectives of PA in childhood are to reduce sedentary lifestyles and promote integral development and the regular practice of PA in a voluntary and deliberate way [4]. Furthermore, regular PA promotes growth and development, as well as contributing to improved physical, mental and psychosocial health [5], which are key aspects for improving physical and cognitive development [6] related to academic performance and psychosocial well-being [7]. In this sense, childhood is a sensitive period for the development of physical and cognitive capacities [4], due to structural and functional changes in brain processes [3], as well as the configuration of behaviour patterns, and the acquisition of healthy habits at this stage is fundamental [8]. Thus, in the last decade, PA in the field of education has become more relevant [9,10], not only due to the fact that it promotes physical well-being but also because of its benefits for cognitive development and academic performance [11]. 

Cognitive functions are highly complex mental capacities related to the acquisition, manipulation and reasoning of information [12] that are conditioned by external factors such as social interaction, socio-economic aspects and environmental stimuli [13]. Although cognitive functions generally improve with age, interindividual differences remain significant throughout development [14]. These differences are influenced by a combination of genetic and environmental factors, including general cognitive ability, motivation, personality traits [15], and perceived academic competence [16,17]. Acknowledging these multifactorial influences is essential to understanding cognitive potential in educational settings [14]. In this context, psychological tests (e.g. Torrance Creative Thinking Test [18] and the School Aptitude Test Level [19]) are often used to estimate aptitude; however, they ultimately reflect performance at the time of assessment [20]. This distinction is important, as test results may be influenced by both stable cognitive traits and situational factors such as motivation, prior knowledge, and test-taking conditions [17]. 

Children differ in the speed and ease with which they acquire new knowledge or skills [17]. These differences often depend on their existing cognitive resources, such as prior knowledge, processing capacity, and learning strategies [16]. As a result, some children grasp complex concepts quickly, while others require more time and support to reach the same level of understanding. This potential, often referred to as aptitude, is distinct from learned abilities and may be shaped by biological traits such as brain structure or developmental timing [21]. 

Creativity is a complex cognitive process and a key attribute for enhancing innovation, problem-solving and the ability to generate original ideas [22]. In particular, divergent thinking is the ability to think of multiple innovative solutions to traditional questions [23]. Therefore, divergent thinking is a predictive factor of human creativity associated with originality, fluidity and flexibility [24]. Consequently, enhancing creativity from childhood represents a competitive advantage for social and human development [25].

On the other hand, intelligence is a mental capacity influenced by multifactorial characteristics that allow human beings to understand, reason, solve problems, plan and think abstractly without prior knowledge (fluid intelligence), as well as to learn from experience and adapt to the environment (crystallised intelligence) [26]. In the academic context, especially in childhood, fluid reasoning is a crucial factor as it is related to inhibitory control, working memory, and the ability to acquire new knowledge quickly and solve problems in complex situations [11]. Consequently, a predictive factor of academic performance is high levels of fluid reasoning, given that these levels are associated with the ability to learn and consolidate new knowledge more efficiently [27]. Thus, creativity and intelligence improve the capacity of students, since they intervene in the teaching and learning process, and are predictors of academic performance and psychosocial adaptation [25,27]. By focusing on creativity and fluid intelligence as expressions of this potential, the study seeks to understand how these cognitive resources may be connected to different aspects of PF in school-aged children. Therefore, understanding the relationship between specific components of PF and cognitive abilities such as creativity and fluid intelligence is crucial in childhood, a stage characterized by rapid neurodevelopment and plasticity [28,29].

In this context, PF has been identified as a powerful biomarker of health [30]; however, secular trends in PF associated with technological development have exhibited a decline in PF and changes in body composition [31,32], as well as in PA levels [33]. This decline in PF represents a major challenge that can compromise not only children’s health but also their motor and cognitive performance [34]. Furthermore, several studies have identified significant differences between boys and girls in CRF, coordination, body composition, and perceived physical competence [35]. Moreover, girls often face additional barriers to PA participation, including lack of time and lower motivation [36,37]. These disparities highlight the need to tailor physical activity strategies to more accurately identify children at risk of poor physical and cognitive outcomes, potentially through sex-specific approaches [35,37]. 

Despite growing interest in the relationship between PF and cognitive functioning, the potential influence of CRF on creativity in childhood remains underexplored. Existing studies have yielded inconsistent findings [38,39], and few have examined how this relationship may vary by age or sex [40,41,42]. This lack of clarity hinders the development of targeted educational or physical activity interventions that could enhance creativity through fitness-related strategies.

A considerable body of research has linked CRF commonly assessed through measures such as VO_2_ peak to improved cognitive outcomes, including attention, memory, and academic performance [40,43]. Although the cognitive benefits of CRF have been well documented, with evidence linking aerobic capacity to improved brain plasticity and cognitive performance [6,10,11,44], fewer studies have examined how other components of PF such as muscular strength, speed, agility, and coordination may contribute to cognitive development in children [11,45,46]. However, the relationship between specific PF components and higher-order cognitive functions, such as creativity and fluid intelligence, remains poorly understood in childhood, with limited and inconsistent findings to date.

Exploring how specific PF components differentially relate to these cognitive abilities could provide a more nuanced understanding of the fitness–cognition relationship. Identifying the most influential domains may also help guide the development of targeted physical activity strategies in educational settings, with the goal of promoting both physical and cognitive development in children. 

To address this gap, the present study adopts a novel and integrative approach that considers the combined contribution of diverse PF components and their relevance to different aspects of cognitive development. To our knowledge, no prior study has jointly examined the associations between multiple components of PF and both creativity and fluid intelligence in childhood. Addressing this gap, the present study aims to determine the relationship between the components of PF, creativity and fluid intelligence, as well as to determine which components of PF are predictors of the analysed cognitive potential. The main hypotheses proposed are as follows: (a) There is a significant association between components of PF (CRF, muscular strength, and speed) and the cognitive domains of creativity and fluid intelligence in children. (b) Specific components of PF will significantly predict creative thinking and fluid intelligence scores in school-aged children.

## 2. Materials and Methods

### 2.1. Design and Participants

This study followed a cross-sectional design. The sample consisted of 584 Spanish schoolchildren aged 6 to 11 years (M = 8.62 ± 1.77), recruited from both urban and rural areas in the autonomous region of Andalusia, located in southern Spain. Children were recruited from different public (n = 5) and private schools (n = 3) in urban and rural areas of Andalusia

The a priori sample size was calculated using G*Power software version 3.1.9.7, assuming a medium effect size (f^2^ = 0.15), an alpha level of 0.05, and a statistical power of 0.80. The analysis indicated a minimum required sample of 400 participants. A stratified convenience sampling method was used, ensuring representation across different age groups (6−11 years), sex, and geographic context (urban and rural schools). Moreover, children provided informed consent forms signed by their parents or legal tutor, which were collected by the teachers and submitted to the researchers. In addition, the respective authorisation was requested from the centres to carry out the research.

The inclusion criteria were (i) neuropsychological impairments; (ii) absence of physical impairments and/or intellectual disability; (iii) health problems (e.g. respiratory or cardiac diseases); and (iv) submission of a signed informed consent form. The study was approved by the Ethics Committee of the University of Jaén (reference code: JUN.21/7.TES). Additionally, participation in the study was entirely free and voluntary. Data anonymity was strictly maintained, with all information handled confidentially and used exclusively for research purposes. A general summary of the study’s findings will be made available to the participating schools upon request. Individual results will remain confidential and will not be shared, in compliance with ethical guidelines. The study was conducted in accordance with the ethical principles outlined in the Declaration of Helsinki [47].

### 2.2. Materials and Procedures

#### 2.2.1. Anthropometric Measures

Height (m) was measured with a stadiometer (Seca 222, Hamburg, Germany), and weight (kg) was measured using an electronic scale (OMRON BF 51). Body mass index (BMI) (kg/m^2^) was calculated using anthropometric measurements by dividing body mass (kg) by the square of body height (m^2^). Waist circumference (WC) was assessed following the International Society for the Advancement of Kinanthropometry guidelines [48]. Umbilical circumference was measured using a non-elastic ergonomic tape measure (Seca 201, Germany; range: 0−150 cm; accuracy: 1 mm).

#### 2.2.2. Fitness Tests

The selected fitness tests were chosen for their validity, reliability, and suitability for school-aged children [49,50,51], as well as their frequent use in studies linking PF to cognitive development [41,52,53].

Sprint speed was assessed via a 25 m sprint in a flat area. The best time was recorded in seconds (s). Two photocells (WITTY, Microgate Srl, Bolzano, Italy) were used to record the time automatically to avoid bias. In addition, each participant performed two trials. 

Handgrip strength was evaluated using a manual dynamometry test. A dynamometer (Takei TKK 5101; Japan) with variable manual adjustment was used; participants performed two trials with each hand and the best trial (kg) was recorded. In addition, the protocol of Ruiz et al. [54], established for the tests, was followed and the final score was obtained from the mean between both hands.

A standing long jump (SLJ) was used for the evaluation of lower-body explosive muscle strength. Participants performed the SLJ from a line marked on the floor, with both feet together, aiming for the maximum possible distance. The SLJ distance was measured from the starting line to the nearest point of contact after landing. Each participant performed two trials, and the greatest distance achieved was recorded (cm).

Aerobic capacity was assessed using the Léger test, also known as the 20 m shuttle run test (20 m SRT) [55]. It consists of a continuous run with progressive intensity. Participants move between point A and point B separated by 20 m consecutively. The test starts with an initial speed of 8.5 km/h and increases by 0.5 km/h per minute (each period). The test ends when the participants do not reach the end of the lines (A or B), according to the sound signals, twice consecutively. The time period during which the participant was eliminated was recorded. The participants’ maximal aerobic performance was recorded to calculate the maximum oxygen consumption (VO_2_ peak). It was calculated using the equation proposed by Léger et al. [55]: VO_2_ peak = (31.025) + (3.238*V) – (3.248*A) + (0.1536 *A*V), where “V” is velocity and “A” is the age of the participant. Furthermore, the level of effort was assessed using the Borg Rating of Perceived Exertion (RPE) scale [56], which ranges from 0 (no exertion) to 10 (maximal effort).

#### 2.2.3. Cognitive Functioning Tests

##### Torrance Creative Thinking Test (TTCT)

The TTCT [18] was used in this study to assess creative potential in children. The TTCT includes two main forms: verbal and figural. In this study, the figural form was used to assess creative thinking. The test comprises three activities, namely picture construction, picture completion and parallel line. The overall assessment time is 30 min, with 10 min allocated for each subtest. With regard to scores, the figural form of the TTCT includes four domains in which fluency (number of ideas produced), flexibility (handling of ideas), originality (how different and out of the ordinary the idea is) and elaboration (development of the idea) are assessed to determine overall creativity. As regards TCTT scores, fluency is scored from 0 to 1 for each relevant answer, with unscored answers excluded from further evaluation. Originality is scored based on the novelty of the participant’s drawings, following Torrance’s Figural Scoring Manual [18]. Listed responses receive 0−5 points depending on rarity; unlisted responses automatically score 5 points. [18]. Title abstraction is scored on a scale of 0 to 3 for abstraction and synthesis [18]. Elaboration counts added details (e.g. embellishments, colours, shading) and assigns scores using manually defined intervals. Flexibility measures the variety of categories used, following Torrance’s manual [18]. A higher score in the TTCT indicates greater creativity [57]. Although the TTCT is often used for identifying gifted individuals, in this study it served exclusively as a measure of creative thinking to examine its association with PF components. In terms of reliability (Cronbach’s α), in the current study, the TTCT demonstrated good reliability (α = 0.816). Previous studies have demonstrated satisfactory results for the overall score (α = 0.720) [58].

##### School Aptitude Test Level 1 (TEA-1)

Children’s learning aptitude was assessed using the School Aptitude Test Level 1 (TEA-1), designed for children aged 8−12 years and administered in the Spanish version [19]. It is a key instrument for assessing intelligence using a classical approach, defined as the ability to learn. The TEA-1 assesses intelligence in three domains: verbal, numerical and reasoning. It is structured in five sections: drawings (15 points), different words (15 points), vocabulary (20 points), reasoning (27 points) and calculations (55 points). The raw score values range between a minimum of 0 and a maximum of 132 points. The total raw score is adjusted for age and then converted into an IQ score, which ranges from 39 to 147 points according to the manual [19]. The TEA-1 demonstrated strong reliability in this study (α = 0.840).

### 2.3. Procedure

The research was carried out in schools during physical education classes. The structured assessment protocol consisted of four sessions. In the first session, the TTCT (three activities) was administered for 30 min (10 min for each activity) followed by the assessment of anthropometric measures. In the second session, two assessment stations were set up for the 20 m SRT and the handgrip strength test. The 20 m SRT assessment was performed in four groups of six participants to avoid research bias. During the third session, the TEA-1 (60 min) was administered, following the guidelines in the manual. Finally, in the fourth session, the 25 m sprint and SLJ test were completed.

Moreover, instructions for the TTCT and TEA-1 activities were carefully explained to the participants and any doubts were resolved. In addition, members of the research group were present at each assessment to answer questions during the evaluation process. For the physical test protocol, a 10 min warm-up was carried out. In addition, the Borg scale was used to assess the perception of effort in the 20 m SRT. 

### 2.4. Statistical Analyses

Data are expressed as means (M) and standard deviation (SD). Normal distribution (Kolmogorov–Smirnov) and homogeneity (Levene) tests were performed prior to analysis. Differences between groups and sex were evaluated using analysis of covariance (ANCOVA) adjusted by age. In regard to the magnitude of the effect, the following criteria were used [59]: 0.00−0.20 (negligible effect); 0.20−0.50 (small effect); 0.50−0.80 (medium effect); and 0.80−1 (large effect). Moreover, a partial correlation analysis (adjusted by sex and age) was performed between creativity and intelligence measures and PF. A linear regression was conducted to explore the predictive ability of PF components (speed, strength and CRF) on creativity and intelligence measures adjusted by sex and age. Prior to conducting the regression analyses, key statistical assumptions were assessed. Multicollinearity was evaluated using the Variance Inflation Factor (VIF), with all values found to be below 5, indicating no significant multicollinearity among predictors. Statistical significance was set at *p* < 0.05 with a CI of 95%. The data were analysed using SPSS v.24.0 for Windows (Chicago, IL, USA). 

## 3. Results

The results of the preliminary analysis of socio-demographic, anthropometric, PF, and cognitive measures by sex are presented in Table 1. Girls showed a significantly higher waist circumference than boys (*p* = 0.012, *d* = 0.193). In terms of PF, boys outperformed girls in all variables: 25m sprint (*p* < 0.001, *d* = 0.740), handgrip strength (*p* < 0.001, *d* = 0.351), SLJ (*p* = 0.032, *d* = 0.154), 20m SRT (*p* < 0.001, *d* = 0.834), and VO_2_ peak. (*p* < 0.001, *d* = 0.390). Regarding cognitive measures, boys scored higher in total creativity score (*p* < 0.001, *d* = 0.149), with significant differences also observed in the domains of originality (*p* = 0.020, *d* = 0.169), elaboration (*p* < 0.001, *d* = 0.599), fluency (*p* < 0.001, *d* = 0.397), and flexibility (*p* < 0.001, *d* = 0.378). In the TEA-1 test, boys scored significantly higher than girls in the total intelligence score (*p* = 0.045, *d* = 0.181) and IQ score (*p* = 0.010, *d* = 0.181).

The relationship between the domains of creativity and intelligence variables and PF performance found positive and negative associations, as shown in Table 2.

In the case of correlation between creativity and PF performance, the findings revealed that better PF was associated with greater creativity (*p* range from = < 0.001 to 0.001). Specifically, greater creativity was associated with shorter execution times (faster time to complete the distance) in the 20 m sprint (r = 0.236, *p* < 0.001).

Likewise, better PF was associated with higher scores in the variables that assess intelligence (*p* range from = < 0.001 to 0.015). The 20 m SRT, which evaluates aerobic capacity, showed the strongest association with the overall intelligence score (r = 0.380, *p* < 0.001), whereas the IQ score showed a positive association with VO_2_ peak.

A linear regression analysis (method-entering) was performed to examine the relationship between the PF components as predictor variables and the overall creativity and intelligence scores (Table 3). The results showed that in the creativity models, the coefficient of determination (*R*^2^ range from = 0.314 to 0.348; <0.001) explained between 31.40% and 36.60% of the variability in the variance, whereas in the intelligence models the variance was explained in between 25.50% and 33.10% (*R*^2^ range from = 0.255 to 0.331; p range from = *p* < 0.001 to 0.001). Overall, these findings suggest that the PF components and age are significant predictors of creative thinking and intelligence.

## 4. Discussion

The aim of this study was to determine the relationship between the components of PF, creativity and fluid intelligence, as well as to determine which components of PF are predictors of the analysed cognitive potential. The main findings of our study indicated that PF components (aerobic capacity, strength and speed) were moderately associated with creativity and fluid intelligence performance. Furthermore, the results of the linear regression model indicated that aerobic capacity (VO_2_ peak), sprint speed (25 m sprint) and SLJ (lower limb strength) showed moderate predictive capacity for creativity and fluid intelligence. However, handgrip strength (upper limb strength) proved to have no predictive value.

These results are in line with previous research [44,60,61,62] that has shown that PF plays a fundamental role in the development, growth and maturation of children due to the profound changes that occur in childhood and their great influence on the holistic development [5]. Likewise, PF, especially CRF, has been consolidated as a predictor of cognitive function and health in children [6,10,11,44]. Although handgrip strength is a widely used indicator of upper−body muscular strength, it did not emerge as a significant predictor of creativity or fluid intelligence in our models. This may be due to the limited relevance of upper-limb strength for the types of cognitive tasks assessed, or to the possibility that handgrip strength does not adequately capture neuromuscular components more directly involved in cognitive processing, such as coordination, agility, or motor control [63]. Additionally, maturational differences among participants may have influenced the results, particularly given the considerable variability in physical development during childhood [64]. Alternatively, the absence of significant associations may reflect a true lack of relationship between upper-body strength and the cognitive outcomes examined [65]. Furthermore, grip strength tasks involve minimal cognitive demand and do not require complex motor coordination, which may limit their relevance to higher-order cognitive functions such as executive functioning or creativity [63,66].

On the other hand, the correlation analyses revealed that children with better PF performance showed significant associations with creativity and fluid intelligence scores, demonstrating that higher levels of PE were associated with better scores in cognitive tests. These findings emphasise the relationship between PF and better cognitive development from childhood. These results are consistent with previous meta-analyses reporting an association with improvements in intelligence [67,68] as well as creativity in children [25,46,69]. This may be explained by the fact that regular practice of PA and improvement in PF are associated with efficiency in the processes of angiogenesis, neurogenesis and synaptogenesis, which are responsible for cerebral vascularisation and the quality of synaptic connections, especially related in the educational context to creativity and fluid intelligence [70,71]. 

A study conducted by Álvarez−Bueno et al. [67] evaluated PF components (strength, speed and CRF) in relation to logical−mathematical intelligence using the Superior Logical Intelligence Test and EVAMAT among 63 schoolchildren, and concluded that SLJ and speed/agility were predictors of logical−mathematical intelligence (*R*^2^ = 0.24 and *R*^2^ = 0.16). Similarly, Bazalo et al. [11] conducted a study with 129 schoolchildren and the results highlighted associations between PF and fluid intelligence. A possible explanation for this could be that PF stimulates the synchrony of neural generators; it helps different areas of the brain work together more efficiently, increasing P3 amplitude (a cerebral signal that appears in attentional processes and information processing) and decreasing P3 latency, indicating faster cognitive processing and more efficient neural communication[28]. Furthermore, around the age of 11 or 12, puberty, related to hormonal increase, significantly affects BDNF regulation, facilitating neuroplasticity and neuronal adaptation due to improved oxygen transport through growth and increased VO_2_ peak [6,72]. In contrast, Fochesatto et al. [44] analysed the association between PF components and fluid intelligence according to body mass index in a sample of 317 schoolchildren and concluded that agility was inversely associated with fluid intelligence only in overweight/obese children (β = −1.506), while CRF and muscular strength showed no association. The authors point out that these discrepancies could be due to the instruments, which may not be sensitive enough to assess cognitive performance (Kaufman Brief Intelligence Test and Raven’s progressive-coloured matrices) or to evaluate PF (20 m SRT and the maximal exercise test using a cycloergometer). A possible explanation regarding the inverse relationship between agility and fluid intelligence in overweight/obese children may be that motor control deficits, reduced PA levels and excess weight can impact on cognitive processing [73,74]. On the other hand, sex differences observed in both PF and cognitive performance, particularly in creativity, may be influenced by a combination of biological and sociocultural factors. Biologically, boys tend to have greater muscle mass and anaerobic capacity during childhood, partly due to differences in hormonal development and maturational timing, which may enhance performance in physical tasks [75,76,77,78]. Sociocultural influences, such as the greater encouragement and opportunity for boys to engage in vigorous physical activity and extracurricular sports, may contribute to the development of motor skills and foster cognitive processes related to creativity, particularly through experiences involving structured physical challenges, problem−solving, and cooperative play [39,79,80]. The literature on sex differences in creativity remains inconsistent. Some studies report higher creative potential in boys [81] whereas others, such as Baer and Kaufman [82] find no significant sex-based differences in either creativity test scores or real-world creative achievements. These contradictions suggest that factors such as socioeconomic status, family environment, and the level of cognitive stimulation at home or school may also influence both physical fitness and creativity outcomes, potentially contributing to the differences observed between boys and girls [83,84]. Similarly, research on intelligence, including fluid intelligence, shows minimal and often non-significant sex differences in childhood. Therefore, any observed disparities should be interpreted cautiously, as they may reflect external variables such as educational context, task format, or assessment methods rather than stable cognitive traits [81,85].

As regards the association between creativity and PF components, our results are consistent with previous findings [25,38,40,86]. This may be explained by the fact that there is a neurobiological relationship between key neurobiological mechanisms and exercise that improves creative thinking through increased blood flow [87]. This promotes cognitive function by increasing levels of acetylcholine, dopamine, BDNF and endocannabinoids, factors responsible for neuronal growth, synaptic plasticity, learning, cognitive flexibility, motivation and learning [25]. For example, a study conducted by Caamaño-Navarrete et al. [41] found a positive association between CRF (measured by the 20 m SRT and VO_2_ uptake) and creativity assessed using the CREA test. The authors pointed out that obese children reported lower creativity than normal-weight children. Similarly, Latorre-Román et al. [38], in a sample of 308 schoolchildren aged between 8 and 12, found that creativity was correlated with PF components: specifically, CRF demonstrated stronger associative power. Also, there is some evidence to suggest that higher PF is associated with better divergent thinking [40].

Some authors have speculated that creativity is a complex psychological process that requires collaboration between different areas of the brain [88], where CRF is key to contributing to greater activation of the cortex and enhanced efficiency of the corresponding cognitive functions responsible for the generation of innovative and original ideas [71,87]. 

Furthermore, the stimulation of cognitive functioning by the increase in blood flow and oxygenation to the brain through exercise enables more efficient neural communication for creativity and intelligence [88,89]. In childhood, the brain increases in size and reorganises structurally, prioritising neural networks over others due to the supply of oxygen (increased VO_2_ uptake), which improves glucose utilisation, favouring cognitive performance [10,90], whereas with age, the brain experiences a reduction in volume and blood flow, which is related to cognitive decline in old age [91]. Therefore, good PF can mitigate the effects of cognitive decline as well as stimulating cognitive functioning [92].

Second, the linear regression analysis reveals that PF components have statistically significantly predicted fluid intelligence and creativity. These findings are consistent with Bazalo et al. [11], who found that speed/agility and medicine ball throws (strength) were predictors of fluid intelligence measured by the WISC−V scale. Similarly, a recent meta-analysis reported small positive effects found in fluid intelligence studies in relation to PF and regular PA practice in promoting beneficial effects on cognition and academic achievement, and in fostering creativity [89]. A probable explanation for this is that the neurobiological influence on cognitive functions due to physical exercise is conditioned by the dose of activity, specifically, the intensity of exercise, the frequency and the duration of the session or practice [93]. Moreover, given that logical−mathematical intelligence is a component of fluid intelligence (reasoning, deduction and abstract thinking), a linear regression analysis indicated that the SLJ test (lower limb strength) is a significant predictor of logical−mathematical intelligence and mathematical competence from an early age (*R*^2^ = 0.24; β = 0.50) [94]. In addition, moderate- to high-intensity exercise stimulates the secretion of dopamine, serotonin and noradrenaline, neurotransmitters that are important for emotional regulation, academic performance and the emotional well-being of children [3]. Furthermore, this finding is line with Latorre-Román, who concluded that CRF was a predictor of creativity in schoolchildren. On the other hand, given that the decline in creativity is believed to begin at the age of five and can reach its peak at the age of eleven, at the end of primary school [95], it is necessary to look for multidisciplinary strategies that contribute to enhancing creativity. Here, it is important to point out that creativity in childhood is the result of the interaction of different contextual, scholastic and environmental factors [13], in which, beyond PA, a holistic vision emphasises the relationship between the factors and their influence on the development of creative thinking [96], as well as the domains of fluid intelligence [13].

Therefore, this study brings a new perspective by exploring the relationship between components of PF, not only CRF, and two complex cognitive functions: creativity and fluid intelligence. Unlike previous research, our study analyses both capacities in the same work, highlighting the relationship between cognitive functioning and a better level of general PF for problem-solving and innovative thinking.

From an educational perspective, the findings suggest that promoting PA that enhance aerobic capacity, muscular strength, and motor skills may contribute not only to physical health but also to cognitive development, particularly in areas such as creative thinking and problem-solving. School-based physical education programs could be designed to include structured, age-appropriate activities that stimulate both PF and cognitive engagement [97]. The observed associations between CRF, strength, and cognitive outcomes such as creativity and fluid intelligence highlight the value of integrating cognitively demanding physical tasks into the curriculum, enhancing a more holistic approach to learning and child development. For example, regular PA, particularly aerobic and coordinative exercises, has been shown to promote brain plasticity, enhance executive functions, and improve information processing speed, all of which are closely related to nonverbal intelligence [40,98]. These benefits may be especially relevant for children with lower verbal fluency, as nonverbal tasks rely more heavily on visuospatial reasoning, attention, and working memory than on language [99]. PA that involves spatial navigation, motor planning, and problem-solving may stimulate neural pathways that support nonverbal cognitive processes, offering alternative routes for cognitive development in children who are less verbally proficient [39,40,97,98].

Although the present study reveals a strong association between PF components and both creativity and fluid intelligence, the findings should be interpreted with caution. The cross-sectional design limits the ability to draw causal conclusions, and the scarcity of prior research exploring these specific cognitive domains in relation to PF further highlights the need for additional studies to confirm and expand upon these results. Nonetheless, the findings emphasize the importance of physical education classes as structured environments for guided PA and support the regular practice of appropriately intense and aerobic-based exercise to promote PF as a potential contributor to cognitive development in children. 

### 4.1. Limitations and Strengths

This study presents several limitations. First, its cross-sectional design precludes causal inference; longitudinal or intervention studies are needed to clarify directionality and underlying mechanisms. Second, unmeasured factors such as extracurricular physical activity, overall activity level (e.g., heart rate data), academic performance, and additional cognitive domains, may have acted as confounders. Despite these constraints, the study’s large sample and its comprehensive analysis of key PF components (aerobic capacity, muscular strength, and speed) in relation to both creativity and fluid intelligence are notable strengths. The holistic perspective adopted connecting distinct fitness domains with complementary cognitive constructs offers an integrated view seldom addressed in the current literature.

### 4.2. Practical Application

These findings have important implications for contributing to closing the gap concerning the relationship among PF components as potential predictors of cognitive functions such as creativity and intelligence. Therefore, schools should implement innovative strategies with a multidisciplinary approach to maximise the benefits of PA for cognitive functioning and academic performance. Also, these findings support the development of educational policies that emphasise the importance of physical education for the physical, cognitive and motor development of children, as well as its relationship with critical stages of development and gaps between the sexes from an early age. Finally, the time allocated to physical education classes is not sufficient to contribute to the comprehensive training and autonomy of the students’ delivered PA practice. Consequently, the family is an important factor in promoting an active lifestyle in children and increasing PF in line with their level of development in an interactive way with cognitive abilities.

## 5. Conclusions

These findings highlight a positive relationship between creativity, fluid intelligence, and components of PF. Notably, strength and CRF showed the strongest associations. Children with higher levels of PF also tended to obtain better scores in creativity. These results suggest that various PF components, not only CRF, may be relevant for enhancing creativity, which is essential for innovation, problem-solving, and the generation of original ideas. However, longitudinal studies are needed to further examine the directionality and stability of these associations over time.

## Figures and Tables

**Table 1 healthcare-13-01682-t001:** Results of sociodemographic, anthropometric, PF and cognitive functioning measures by sex.

Variables	ALL Mean (SD)	Girls Mean (SD)	Boys Mean (SD)	*p*-Value	*d Cohen’s*
	(n = 584)	(n = 349)	(n = 235)		
Age (years)	8.62 (1.77)	8.43 (1.77)	8.93 (1.72)	<0.001	0.286
Anthropometric variables					
BMI (kg/kg/m^2^)	18.43 (3.84)	18.56 (3.87)	18.36 (3.78)	0.467	0.052
Waist circumference (cm)	62.74 (10.93)	64.15 (11.37)	62.01 (10.63)	0.012	0.193
Fitness variables					
25m Sprint (s)	5.64 (0.79)	5.84 (0.81)	5.29 (0.63)	<0.001	0.740
Handgrip strength (kg)	13.61 (5.36)	12.93 (4.92)	14.80 (5.87)	<0.001	0.351
SLJ (cm)	114.77 (37.28)	112.66 (34.94)	118.45 (40.84)	0.032	0.154
20 m SRT (periods)	2.70 (1.82)	2.19 (1.50)	3.62 (1.99)	<0.001	0.834
VO_2_ peak (mL·${kg}^{-1}$ 1·${min}^{-1}$)	46.25 (4.14)	45.68 (3.78)	47.28 (4.54)	<0.001	0.390
Borg scale (0−10 pts)	7.48 (2.13)	7.31 (2.32)	7.81 (1.69)	<0.002	0.239
Cognitive measures					
Creativity (TTCT)					
Originality (3−175 pts)	66.60 (32.62)	64.51 (33.72)	70.01 (30.51)	0.020	0.169
Elaboration (0−34 pts)	30.19 (20.04)	23.41 (15.67)	34.97 (21.38)	<0.001	0.599
Fluency (0−40 pts)	18.17 (8.24)	16.93 (8.79)	20.14 (6.85)	<0.001	0.397
Flexibility (0−29 pts)	14.35 (6.43)	13.44 (6.96)	15.82 (5.13)	<0.001	0.378
Total creativity score (0−269)	124.64 (51.22)	121.82 (54.62)	129.40 (44.60)	<0.001	0.149
Intelligence (TEA-1)					
Verbal (0−50 pts)	29.01 (9.02)	28.59 (8.98)	29.61 (9.06)	0.145	0.113
Reasoning (0−27 pts)	15.90 (5.50)	15.90 (5.43)	16.02 (5.62)	0.673	0.021
Numerical (0−55 pts)	11.89 (19.85)	11.06 (19.66)	13.15 (20.13)	0.063	0.105
Intelligence score (0−132 pts)	56.37 (24.18)	55.11 (23.77)	58.26 (24.72)	0.045	0.130
IQ score (39−147 pts)	94.95 (19.91)	93.51 (20.47)	97.11 (18.98)	0.010	0.181

BMI = body max index; SLJ = standing long jump; 20 m SRT = shuttle run test; VO_2_ peak = maximum oxygen uptake; TTCT = Torrance creative thinking test; TEA-1 = school aptitude test; IQ = intelligence quotient.

**Table 2 healthcare-13-01682-t002:** Association between cognitive functioning measures with PF.

Variables	PF Variables				
Creativity Variables (TTCT)	25m Sprint (r)	Handgrip Strength (r)	SLJ (r)	20 m SRT (r)	VO_2_ (r)
Originality	−0.309 ***	0.258 ***	0.163 ***	0.205 ***	
Elaboration	−0.130 **		0.142 **	0.260 ***	0.234 ***
Fluency	−0.371 ***	0.273 ***		0.279 ***	0.111 **
Flexibility	−0.307 ***	0.241 **	0.116 **	0.243 ***	
Total creativity score	−0.236 ***	0.232 ***	0.203 ***		0.129 **
Intelligence variables (TEA−1)					
Verbal aptitude	−0.171 ***	0.273 ***	0.182 ***	0.169 ***	
Reasoning	−0.129 **	0.236 ***	0.111 **	0.262 ***	0.158 **
Numerical	−0.183 ***	0.146 **		0.322 ***	0.101 *
Intelligence score	−0.259 ***	0.263 ***		0.380 ***	0.142 **
IQ score	−0.182 ***			0.247 ***	0.118 *

r = Pearson’s correlation coefficient; SLJ = standing long jump; 20 m SRT = shuttle run test; VO_2_ peak=maximum oxygen uptake; TTCT = Torrance creative thinking test; TEA−1= school aptitude test; IQ= intelligence quotient; * *p* < 0.05, ** *p* < 0.01, *** *p* < 0.001. Descriptive statistics for the main variables are based on n = 584 children.

**Table 3 healthcare-13-01682-t003:** Linear regression analysis between the cognitive functioning measures with PF.

Variables	*R*^2^ (*p*-Value)	Predictors Variables	ꞵ	*p*-Value	IC (95%)	
Total Creativity score	0.339 (<0.001)	25 m Sprint	−0.126	0.018	−11.612	−1.100
	0.348 (<0.001)	SLJ	0.168	0.011	0.139	0.397
	0.319 (<0.001)	20 m SRT	0.123	0.031	0.326	6.975
	0.314 (<0.001)	VO_2_ peak	0.203	<0.001	3.807	1.327
Intelligence score	0.331 (<0.001)	25 m Sprint	−0.186	0.001	−10.897	−3.982
	0.303 (<0.001)	SLJ	0.929	0.001	1.250	0.310
	0.255 (<0.001)	20 m SRT	0.312	<0.001	5.694	9.830
	0.307 (<0.001)	VO_2_ peak	0.484	<0.001	3.382	1.762

Note: SLJ = standing long jump; 20 m SRT = shuttle run test; VO_2_ peak = maximum oxygen uptake.

## Data Availability

The data that support the findings of this study are available from the corresponding author, upon reasonable request.

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
