# Peer review of "Can the Components of Physical Fitness Be Linked to Creative Thinking and Fluid Intelligence in Spanish Schoolchildren?"

_healthcare, 2025, doi:10.3390/healthcare13141682_

Round 1
Reviewer 1 Report
Comments and Suggestions for Authors
First of all, congratulations on this article.
- Introduction
Strengths: The introduction provides a comprehensive background on the importance of physical activity (PA) and physical fitness (PF) in childhood, linking them to cognitive development, creativity, and fluid intelligence. The references are relevant and support the study's rationale.
Suggestions: Clarify the gap in the literature more explicitly. While the authors mention the scarcity of studies on PF components beyond cardiorespiratory fitness (CRF), a stronger statement about the novelty of investigating creativity and fluid intelligence together would help.
2. Research Design
Strengths: The cross-sectional design is appropriate for exploring associations between PF components and cognitive outcomes. The sample size is adequately justified and sufficiently large.
Suggestions: Briefly address potential limitations of the cross-sectional design (e.g., inability to infer causality) in the methods or discussion.
3. Methods
Strengths: The methods are well-detailed, including clear descriptions of fitness tests (e.g., 20 m SRT, handgrip strength) and cognitive assessments (TTCT, TEA-1). The inclusion/exclusion criteria are appropriate.
4. Results
Strengths: Results are clearly presented in tables and text, with appropriate statistical analyses (ANCOVA, regression). Sex differences in PF and cognitive scores are highlighted effectively.
Suggestions: Include effect sizes (e.g., Cohen’s *d*) for all significant findings to aid interpretation.
5. Discussion
Strengths: The discussion contextualizes the findings well, linking them to prior research on PF, BDNF, and cognitive function. The practical implications for schools are valuable.
Suggestions: Address the lack of predictive value for handgrip strength more thoroughly. Is this due to measurement limitations or a true lack of association?
Expand on the sex differences observed. Why might boys outperform girls in both PF and creativity? Could societal or biological factors play a role?
6. Tables and Figures
Strengths: Tables are clear and well-organized, with appropriate labels and statistics.
Author Response
We would like to thank the time spent reviewing our manuscript. We have considered all suggestions and we believe our manuscript is stronger as a result of the changes that we have introduced in the revised version of the manuscript. Changes to the original manuscript are highlighted in yellow font, and an itemized point-by-point response to your comments in this document.

Reviewer 2 Report
Comments and Suggestions for Authors
The reason for studying the relationship between physical fitness (PF) components and creativity and fluid intelligence should be explained in more detail, in order to determine which PF components are predictors of the cognitive potential analyzed in children. And in any case, the educational implications of working on the components that enhance cognitive development should be explained at the end.
The Torrance Creative Thinking Tests (TTCT) are designed to identify and assess creative potential using two parts: a verbal test and a figurative test.
This should be explained in more detail in the study. This test is widely used to identify gifted students, so it should be reflected in the results. That is, the number of potentially gifted students in light of the results.
On the other hand, some authors understand academic aptitudes as individual differences among individuals, as individuals have different levels of development, especially at the educational stage in which the research is conducted. These differences may be due to personal or hereditary factors, or environmental and contextual factors. Other authors emphasize a performance index because they consider that if aptitudes are to be assessed with psychological tests, the result of said measurement will be the level of performance achieved and not the measurement of an aptitude factor, whether hereditary or environmental, alone. From this, aptitudes are capacities that allow individuals to acquire efficiency and speed in the execution of activities. Other authors also seek to establish conceptual differences between aptitude and ability and emphasize that aptitude is associated with anatomical particularities that form innate differences, in this case, of children. These aspects should be addressed in the study.
The authors handle a total of 584 Spanish schoolchildren 116 (6–11 years old; age = 8.62 ± 1.77 years old) were recruited from urban or rural areas of 117 Andalusia to take part in this study. They should explain the context, the region, in more detail. And on the other hand, they have not considered the center variable, the context variable, since there are already quite a few studies that are determining how the school and family environment determines both the physical condition and other aspects of development. But in any case it was not done because it is not requested now.
The reason for selecting fitness tests should be explained.
The educational implications of the research should be explained in more detail.
How physical activity can increase the nonverbal intelligence of children who are not fluent should be explained in more detail.
Author Response

(The authors gave the same response as above.)

Reviewer 3 Report
Comments and Suggestions for Authors
I am pleased to have had the opportunity to review the paper titled Can the components of physical fitness be linked to creative thinking and fluid intelligence in childhood?
First, I would like to compliment the authors for choosing a topic that, as they have pointed out, has been insufficiently explored in previous research. Although considerable attention has been given to physical fitness (PF) and physical activity (PA) and their relationship with various developmental domains, their connection to creativity and fluid intelligence remains underexplored.
The title corresponds well with the content of the paper.
In the introduction, the authors systematically present the role of PA in promoting growth and development, as well as in improving physical, mental, and psychosocial health. They also emphasize the importance of the early developmental period for cognitive functioning. Furthermore, they list the PF and PA factors that contribute to development and how they influence health and general cognitive functioning.
The introduction is well written and supported by key literature data.
The methodology is clearly described, allowing for the reproducibility of the study. The process of participant recruitment, inclusion criteria, and ethical standards were clearly explained. The description and selection of instruments are appropriate, with very precise explanations of their characteristics, administration, and interpretation.The testing procedure is also described in a way that allows for clarity and reproducibility.
The results are clearly presented through three tables. The applied statistical analysis, which includes both descriptive and inferential statistics, is adequate.
Ethical committee approval was obtained.
The methodology corresponds well to the research design.
In the discussion section, the obtained results are adequately interpreted, discussed, and connected with relevant literature. Data from studies with conflicting results as well as those that support the obtained data are mentioned.
Author Response

(The authors gave the same response as above.)

Reviewer 4 Report
Comments and Suggestions for Authors
Dear authors, I appreciate the opportunity to review the manuscript.
Regarding the title, it is clear and generally reflects the purpose of the study, although it could benefit from incorporating more detail about the sample, as this would help to better contextualize the study.
The abstract is well-structured and provides an initial contextual line that clearly frames the reader, addressing the study's objectives, methodology, and results. In this way, the reader can follow the argumentative thread of the study. However, the language used is somewhat technical for all types of audiences, which may limit its reach. Additionally, it is suggested to incorporate the study's relevance for the educational context into the abstract.
The objectives are clearly stated, although a hypothesis is not explicitly formulated, which could be relevant.
The introduction gathers relevant and recent background information on the relationship between physical activity, cognitive development, creativity, and fluid intelligence. However, there is a certain level of redundancy and reiteration of concepts, which makes it difficult to maintain focus while reading. I also believe that the research problem could be defined with greater precision from the outset.
The theoretical framework is well-founded and supported by recent bibliography (last five years). However, its structure could be improved if organized by dimensions, for example, physiological mechanisms (BDNF, neurotransmitters), general cognitive benefits, and empirical evidence, which would facilitate reading. I also recommend explicitly stating the existing gaps in the literature that this study aims to address.
Regarding the methodological section, it is well-structured, clearly explains the study design and sample, and uses validated instruments to evaluate creativity (TTCT) and intelligence (TEA-1), in addition to standardized physical tests. However, some important aspects are not detailed, such as the parameters of the sample size or the procedures to ensure quality in test application. It would also be relevant to include information on the sociodemographic characteristics of the participants.
The study has institutional ethical approval, although it is not specified from which university, and informed consent was obtained from the legal representatives of the participants. However, it is not mentioned whether results were returned to the participants or how this was carried out.
The results are well-structured and presented clearly to the reader. They include descriptive analyses, correlations, and regressions, with an indication of effect sizes (R2). Additionally, variables such as gender and age are considered. However, some sections are redundant and could be summarized. It is also not specified whether a statistical assumption evaluation, such as multicollinearity in regression models, was performed.
Regarding the tables, they could be improved in various aspects, such as in table one, where the score range for some variables is not explained, in table two, the sample size is missing in the legend, and in table three, it could be better organized for improved interpretation.
Regarding the discussion section, the main findings are correctly interpreted and linked to existing literature, integrating relevant physiological and contextual explanations. Even so, there is a lack of critical analysis of possible biases, a reflection on reverse causality, and a more detailed discussion on the limitations of the cross-sectional design and possible confounding variables (such as nutrition, sleep, or cognitive stimulation at home).
In the case of the conclusions, they adequately align with the findings and highlight their relevance for the educational field. However, it is recommended to avoid statements that suggest direct causality.
Regarding the bibliography, most of it is up-to-date, including bibliography from the last 5 years.
Author Response

(The authors gave the same response as above.)

Round 2
Reviewer 4 Report
Comments and Suggestions for Authors
Dear authors,
According to the revisions made by the reviewers, most of the observations were satisfactorily incorporated, with positive advancements in terms of the title, abstract, theoretical framework, methodology, tables, and results, as well as the addition of hypotheses and evaluation of statistical assumptions. However, there are some minor aspects that need to be reviewed, such as greater precision in defining the problem and specifying whether the results will be returned to the participants and how this will be guaranteed.
I thank you again for your work and I am sure that progress on these last points will significantly improve the manuscript.
Author Response
We would like to express our sincere gratitude for the time and effort dedicated to reviewing our manuscript. We have carefully considered all the suggestions provided and have incorporated several revisions in response to your valuable comments. We believe these modifications have significantly improved the quality and clarity of the manuscript. All changes have been highlighted in yellow in the revised version of the manuscript.